# Inequalities in Mortality and Access to Hospital Care for Cervical Cancer—An Ecological Study

**DOI:** 10.3390/ijerph182010966

**Published:** 2021-10-19

**Authors:** Luiz Vinicius de Alcantara Sousa, Erika da Silva Maciel, Laércio da Silva Paiva, Stefanie de Sousa Antunes Alcantara, Vânia Barbosa do Nascimento, Fernando Luiz Affonso Fonseca, Fernando Adami

**Affiliations:** 1Laboratory of Epidemiology and Data Analysis, FMABC University Center, Santo André 09060-590, Brazil; laercio.paiva@fmabc.br (L.d.S.P.); adamifernando@uol.com.br (F.A.); 2Graduate Program in Science and Health Education PPGECS_UFT, Federal University of Tocantins (UFT), Miracema do Tocantins 77650-000, Brazil; erikasmaciel@uft.edu.br; 3FMABC University Center, ABC Medical School, Santo André 09060-590, Brazil; antunestefanie@gmail.com (S.d.S.A.A.); vania.nascimento@fmabc.br (V.B.d.N.); 4Clinical Analysis Laboratory, FMABC University Center, Santo André 09060-590, Brazil; profferfonseca@gmail.com

**Keywords:** neoplasms of the cervix, mortality, length of hospitalization, epidemiology

## Abstract

Cervical cancer is the second most common form of cancer in the world among women, and it is estimated to be the third most frequent cancer in Brazil, as well as the fourth leading cause of death from cancer. There is a difference in cervical cancer mortality rates among different administrative regions in Brazil along with an inadequate distribution of cancer centers in certain Brazilian regions. Herein, we analyze the trends in hospital admission and mortality rates for CC between 2000 and 2012. This population-based ecological study evaluated the temporal trend in cervical cancer between the years 2000 and 2012, stratifying by Brazilian administrative regions. The North and Northeast regions had no reduction in mortality in all age groups studied (25 to 64 years); when analyzing hospitalization rates, only the age group of 50 to 64 years from the North Region did not present a reduction. During the years studied, in the South Region, the age group ranging from 50 to 54 years had the greatest reduction in mortality rates (β = −0.59, *p* = 0.001, r^2^ = 0.63), and the group ranging from 45 to 49 years had the greatest reduction in hospital admission rates (β = −8.87, *p* = 0.025, r^2^ = 0.37). Between the years 2000 and 2012, the greatest reduction in the incidence of UCC was in the South Region (β = −1.43, *p* = 0.236, r^2^ = 0.12) followed by the Central-West (β = −1, *p* < 0.001, r^2^ = 0.84), the Southeast (β = −0.95, *p* < 0.001, r^2^ = 0.88), the Northeast (β = −0.67, *p* = 0.080, r^2^ = 0.25), and, finally, by the North (β = −0.42, *p* = 0.157, r^2^ = 0.17). There was a greater reduction in mortality rates and global hospitalization rates for CC in Brazil than in the United States during the same period with exceptions only in Brazil’s North and Northeast regions.

## 1. Background

Cervical cancer (CC) is the second most common form of cancer in women worldwide, and almost 80% of cases occur in developing countries [1]. It is estimated that cervical cancer is the third most frequent type of tumor occurring in the female population behind only breast and rectal cancer, and it is the fourth leading cause of cancer death in women [2].

The mortality rate from cervical cancer in Brazil is high, constituting a serious public health problem. According to the Ministry of Health, in the historical series ranging from 1979 to 2005, age-adjusted mortality rates rose from 4.97 to 5.29 per 100,000 women, representing an increase of 6.4% over 26 years [3,4,5].

A survey that took place between 1995 and 2002 involving 89 hospitals and 7 isolated chemotherapy or radiotherapy services linked to a High Complexity in Oncology Center (CACON) found that 45.5% of the women studied already had stage III or IV cancer of the cervix at the time of diagnosis and concluded that there were problems in access to services as well as limitations to the information system, revealing, above all, a lack of oncology centers outside the country’s large capitals [3,6,7].

Cervical cancer is a neoplasm with high incidence and mortality rates that is one of the biggest causes of female morbidity in Brazil. This study aims to analyze trends in mortality and hospital admissions rates for CC between 2000 and 2012.

## 2. Method

This is an ecological, population-based study that uses Brazilian data to evaluate temporal trends in cervical cancer between the years 2000 and 2012. Brazilian administrative regions were chosen as study areas in order to ensure more complete data.

### 2.1. Data Source

The survey is composed of all deaths from cervical cancer registered by the Cervical Cancer Information System (SIM) in women between 25 and 64 years of age, the age group in which a cytopathological exam (Pap smear) is offered to Brazilian women. The Pap smear is used for screening and diagnosis of CC [5,6]. The specific period studied ranged from 1 January 2000, to 31 December 2012. Data from the Department of Informatics of the Unified Health System (DATASUS—www.datasus.gov.br (accessed on 12 April 2019)), a free access database that represents the main source of health information in the country, provides health information for states, municipalities, and the Federal District [4].

The reliability of the SIM data can be analyzed through the quality of information, as well as the territorial coverage of the system. Its quality can be validated using its reported proportion of deaths through poorly defined causes, which was approximately 6% [8] in 2011 with coverage of 96.1% [9].

The distributions of cervical cancer admissions and mortality rates were coded according to the 10th International Classification of Diseases (ICD) as C53—malignant neoplasm of the cervix based on the topographic locations of the uterine lesions analyzed [10].

### 2.2. Procedure for Collecting Data

#### Deaths from Cervical Cancer

The collection of information on deaths by CC followed these steps:
-Access to mortality data was based on the following sequence within the DATASUS system:Vital statistics;Mortality between 2000 and 2012;General mortality;Geographical coverage.-The deaths related to each code were extracted and stratified according to the following variables:Age group (from 25 to 64 years, divided into 5-year ranges);Location (administrative regions);Year (2000 to 2012).

### 2.3. Resident Population

The censuses and intercensal projections made available by the Brazilian Institute of Geography and Statistics (IBGE) on the DAASUS website were utilized in order to collect population data, following this sequence within the DATASUS system:Demographic and socioeconomic data;Resident population data;Censuses (1980, 1991, 2000, and 2010), Count (1996), and intercensorial projections (1981 to 2012) according to age, sex, and domiciliary situation;Geographical coverage.

### 2.4. Cervical Cancer Mortality

Gross mortality rates were calculated for 100,000 women, by age group and year, for the five Brazilian regions (North, Northeast, Southeast, Center-West and South) during the study period. At the end of the gross rate estimation, based on the age distribution of the World Health Organization population, mortality went through standardization by age using the direct method [11].

### 2.5. Hospital Admissions for Cervical Cancer

Hospitalizations are presented according to the technical norms of the lists of tabulations for morbidity that are available in the Hospital Information System (SIH).

The SIH provides demographic and clinical data from the Hospital Inpatient Authorization (IAH), allowing hospital morbidity and mortality to be defined within the scope of the SUS’s services [12]; however, hospitalizations of the population using private health insurance are not included.

Information on CC hospitalizations was stratified according to variables:Region;Morbidity list code CID-10 (malignant neoplasm of the cervix);Age group.

### 2.6. Data Analysis

Regression models were used for a statistical analysis evaluating mortality for this neoplasia with dependent variables of hospitalization and mortality rates for CC (dependent variables, Y) and the independent variable of time (independent variable, X) during the study period (2000 to 2012).

The research also estimated the standardized trend for each location and age group with a confidence level of 95% using the statistical program Data Analysis and Statistical Software for Professionals (Stata) Version 16.0^®^.

## 3. Results

In Brazil, between 2000 and 2012, there were 37,892 deaths and 286,975 hospitalizations caused by cervical cancer in women aged 25 to 64 years. There were differences among the age groups and the regions studied with 14,065 hospital admissions in the North, 74,705 in the Northeast, 106,565 in the Southeast, 70,465 in the South, and 21,175 in the Midwest (Table 1).

The risk of death and the incidence of hospital admissions (per 100,000 women) for CC tended to increase with increased age, particularly in the age groups ranging from 45 to 49 (770 deaths) and 40 to 44 years (2909 admissions) in the North Region; in the age ranges of 45 to 49 (1783 deaths) and 40 to 44 years (16,134 hospitalizations) in the Northeast; in the 50–54 (2576 deaths) and 45–49 year (21,140 admissions) age ranges in the Southeast; and 546 deaths between the ages of 45 and 49 years as well as 4706 hospitalizations in the 40–44 age group in the Southwest Region (Table 1).

The North and Northeast regions had no reductions in mortality in any of the age groups studied (25 to 64 years); however, in the North, only the group ranging from 50 to 64 years showed no reduction in hospitalization rate. In the Southern region, the greatest reduction in mortality rate occurred in the age group ranging from 50 to 54 (β = −0.59, *p* = 0.001, r^2^ = 0.63), and the group ranging from 45 to 49 years experienced the highest reduction in hospital admission rate (β = −8.87, *p* = 0.025, r^2^ = 0.37) (Table 2).

Variations in mortality and hospitalization rates by Brazilian region can be visualized in Figure 1 and Figure 2, which present the quartiles for the observed periods. It was observed that only the North and the Northeast experienced increases in mortality (β = 0.18, *p* = 0.008, r^2^ = 0.48; β = 0.18, *p* = 0.002, r^2^ = 0.48, respectively), and the other regions presented reduced rates: Southeast (β = −0.07, *p* < 0.001, r^2^ = 0.79), South (β = −0.11, *p* < 0.001, r^2^ = 0.79), and Central-West (β = −0.03, *p* = 0.278, r^2^ = 0.1) (Figure 3). Between 2000 and 2012, the greatest reduction in the incidence of UCC occurred in the South Region (β = −1.43, *p* = 0.236, r^2^ = 0.12) followed by the Central-West (β = −1.01, *p* < 0.002, r^2^ = 0.59), the Southeast (β = −0.95, *p* < 0.001, r^2^ =0.88), the Northeast (β = −0.67, *p* = 0.080, r^2^ = 0.25), and, finally, the North (β = −0.42, *p* = 0.157, r^2^ =0.17) (Figure 4).

The age-standardized CC mortality rate in Brazil varied widely, but it decreased at the end of the time period ranging from 2000 to 2012 (β = −0.02, *p* = 0.065, r^2^ = 0.27); the rate of hospitalizations 100,000 women) in the study period (β = −1.01, *p* = 0.002, r^2^ = 0.59) also decreased from 26.31 (95% CI: 26.30, 26.32) to 18.48 (95% CI: 18.48, 18.48) (Table 3).

## 4. Discussion

In this study, there were discrepancies in CC hospital admissions among the age groups and the federative regions of Brazil studied with higher numbers of hospitalizations in the Northeast and Southeast.

The highest death rates and numbers of hospitalizations were concentrated in the 40-to-49-year-old age group in all of Brazil’s federative regions. The Southeast had higher numbers of CC deaths and hospitalizations than other regions [10,11,12,13].

CC mortality and the incidence of CC hospitalizations in Brazil decreased overall during the years studied. The South experienced the greatest decreases in mortality and incidence of hospitalization, but the North and Northeast regions had no reductions in rates.

In Brazil, inequality in access to health services is still frequently discussed due to its influence on the high incidence, diagnosis in more advanced stages, and high mortality rate of CC. This supports the theory that even though the number of preventive exams has increased in recent years, access to health services is still related to socioeconomic levels within the country [14,15].

Cervical cancer is an avoidable disease that is sensitive to primary prevention actions (vaccine for HPV as well as condoms distributed free of charge) in addition to the early detection of precursor lesions through examination using oncocytic colpotiology of the cervix and treatments available in Brazilian Unified Health System–SUS territories. This has led to cervical cancer survival in almost all cases diagnosed early [10,11,12,13].

In the historical series ranging from 1996 to 2010, 89,764 cervical cancer deaths were recorded in Brazil, and mortality declined from 8.04 (100,000 inhabitants) in 1996 to 6.36 (100,000 inhabitants) in 2010. Among regions, the reduction was significant in the Central West (1.3% per year), Southeast (3.3%), and South (3.9%), with mortality stability maintained in the North Region and an increase only in the Northeast [15,16].

The National Cancer Institute (INCA) has reported that the estimated incidence of CC in Brazil is distributed among the regions as follows: North (23.97%), Midwest (20.72%), Northeast (19.49%), Southeast (11.30%), and South (15.17%). Other studies have found declines in deaths from cervical cancer in the Southeast, South, and Central-West regions; this did not occur in the North and Northeast regions, which witnessed increases in mortality from cervical cancer in the 1980s and 2010 in Brazil [17].

Higher admission values for CC were found in the less-developed North and Northeast regions of the country, which showed similar results as in a survey of low-income black women in populous neighborhoods of Salvador. This shows that low socioeconomic level is considered a risk factor for the development of CC, and that difficulty in accessing health services is an important variable in both morbidity and mortality [18,19].

The technological advances in primary and secondary prevention that have been instituted in the national territory since the year 2000 have led to reductions in hospitalization and mortality rates in the South and Southeast regions [20,21]. Nonetheless, in Brazil, availability of and access to health systems have been widely studied and have been identified as limitations for the control of CC [22,23].

These results follow the distribution of the incidence of cervical cancer in less developed countries where the risk is about twice as high as in more developed countries, having increased significantly in the age group ranging from 45 to 49 years [5].

However, another survey from Ghana, which described the basic characteristics of women diagnosed with CC as well as diagnostic and treatment methods, concluded that increased risk was present among women in the 30-to-39-years age group [24,25].

In the city of São Gonçalo in Rio de Janeiro, in 2005, CC was the second most frequent cause of death due to neoplasia in women, showing a significant upward trend in mortality between 1980 and 2005 in the populations ranging from 20 to 29 (β= 0.180), 60 to 69 (β = 0.013), and 70 and more years (β = 2.22) [26].

The Brazilian territory is marked by deep regional inequalities that directly affect the supply of both low complexity and high complexity medical care services, creating a problem among administrative regions [27]. The current study corroborates the findings of other authors who have underscored national disparities in both overall mortality and CC mortality with trends toward reduction and/or stability in more developed regions and increases in less developed regions [28,29].

A database study showed a reduction in coefficients of mortality from CC among female residents of the city of Recife, the capital of the federal state of Pernambuco in Northeast Brazil. Our results demonstrate an increase in the mortality rate from CC, which is discordant because our number of deaths is from a wider geographical and population coverage [30].

The DATASUS database is a national registry from the Ministry of Health in Brazil; however, gaps in its information and underreporting of data are limitations in our study.

The novelty of this study is in its presentation of trends in hospital admissions due to cervical cancer within high-complexity centers of the Brazilian single health system as well as in its presentation of discrepancies, showing that free access to prevention and treatment of this disease continues to be influenced by social, educational, and economic determinants. It also warns that services must be organized and directed to vulnerable populations and regions in order to reduce the morbidity and mortality that are dependent on the early diagnosis of cervical cancer.

National data identify regional inequalities linked to factors including geographical barriers, the bureaucracy of effective consultations and care, lack of knowledge about the reason for examination, a lack of knowledge about the relationship between cancer and death, and events that make early detection, which considerably reduces female morbidity and mortality, impossible [31].

The current study observed that cervical cancer is present in all age groups studied across all regions, and it is still responsible for many female deaths in Brazil. Therefore, it is necessary to obtain precise knowledge about the factors leading to worse results in less developed regions and to continue to develop strategies advocating for more information and dissemination of knowledge so that more women are aware of the importance of cancer prevention.

## 5. Conclusions

There was an overall reduction in CC mortality rates in Brazil with exceptions in the North and Northeast regions. There was no reduction in the Brazilian CC hospitalization rate.

## Figures and Tables

**Figure 1 ijerph-18-10966-f001:**
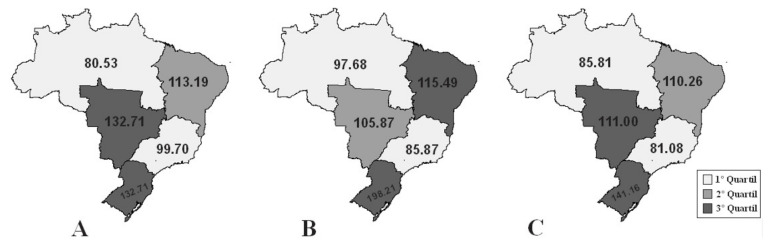
Quartiles of hospitalization rates for cervical cancer in the Brazilian regions in the periods observed. (**A**) 2000–2003; (**B**) 2004–2007; (**C**) 2008–2012.

**Figure 2 ijerph-18-10966-f002:**
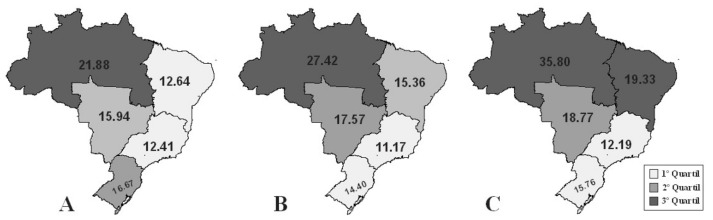
Quartiles of cervical cancer mortality rates in the Brazilian regions in the periods observed. (**A**) 2000–2003; (**B**) 2004–2007; (**C**) 2008–2012.

**Figure 3 ijerph-18-10966-f003:**
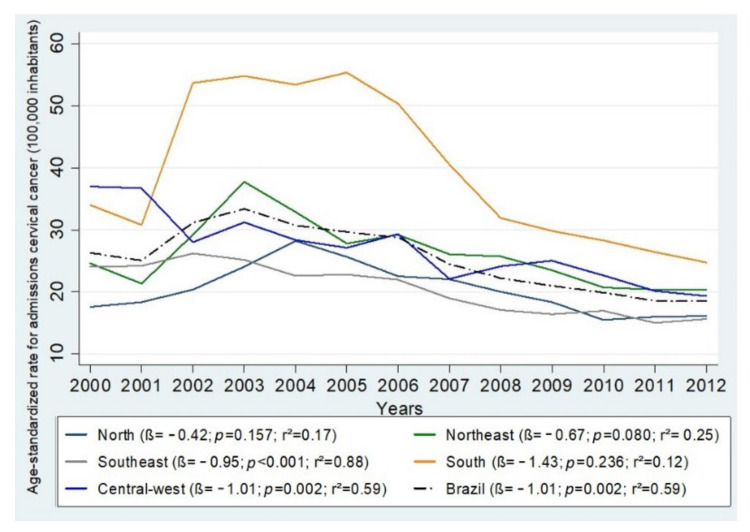
Incidence of hospital admissions for cervical cancer (100,000 inhabitants), resident in Brazil aged 25–64 years, in the period from 2000 to 2012 and estimates obtained from linear regression, according to year and regions.

**Figure 4 ijerph-18-10966-f004:**
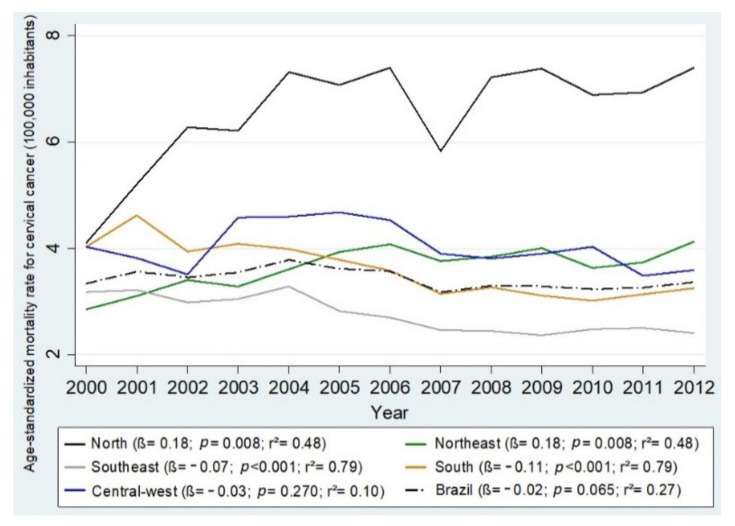
Mortality rate for cervical cancer (100,000 inhabitants), resident in Brazil aged 25–64 years, in the period from 2000 to 2012 and estimates obtained from linear regression, according to year and regions.

**Table 1 ijerph-18-10966-t001:** Loading cervical cancer † in Brazilian women aged 25 to 64 during.

Country Regions	Deaths (*n*)	Age-Standardized Mortality (95% CI)	Hospital Admissions	Age-Standardized Admissions (95% CI)	Proportional Mortality (95% CI)
**Age group (years) ***					
**North**					
25–29	177	2.09 (2.08 to 2.09)	797	9.45 (9.44 to 9.46)	3.55 (3.46 to 3.64)
30–34	375	5.22 (5.21 to 5.23)	1587	22.10 (22.08 to 22.11)	7.12 (6.94 to 7.29)
35–39	496	8.24 (8.23 to 8.24)	2299	38.21 (38.18 to 38.25)	8.89 (8.68 to 9.10)
40–44	677	13.59 (13.58 to 13.60)	2909	58.42 (58.37 to 58.48)	10.61 (10.38 to 10.84)
45–49	770	19.24 (19.23 to 19.26)	2616	65.36 (65.29 to 65.42)	9.61 (9.42 to 9.80)
50–54	704	22.62 (22.60 to 22.64)	1718	55.22 (55.15 to 55.29)	7.61 (7.48 to 7.75)
55–59	617	25.37 (25.34 to 25.40)	1221	50.22 (50.15 to 50.29)	6.19 (6.08 to 6.30)
60–64	529	28.47 (28.42 to 28.51)	918	49.41 (49.34 to 49.49)	4.92 (4.84 to 5.01)
**Northeast**					
25–29	347	1.21 (1.21 to 2.65)	3595	12.55 (12.54 to 12.56)	2.59 (2.55 to 2.62)
30–34	679	2.66 (2.65 to 2.66)	7206	28.23 (28.21 to 28.24)	4.43 (4.37 to 4.49)
35–39	1083	4.81 (4.80 to 4.82)	12,255	54.50 (54.48 to 54.52)	5.85 (5.78 to 5.93)
40–44	1422	7.22 (7.21 to 7.22)	16,134	82.02 (81.98 to 82.05)	6.00 (5.93 to 6.07)
45–49	1783	10.63 (10.62 to 10.64)	15,222	90.81 (90.77 to 90.85)	5.96 (5.90 to 6.01)
50–54	1719	12.28 (12.28 to 12.29)	9402	67.18 (67.14 to 67.22)	4.91 (4.86 to 4.96)
55–59	1683	14.54 (14.53 to 14.55)	6004	51.88 (51.86 to 51.90)	4.31 (4.27 to 4.34)
60–64	1549	15.73 (15.72 to 15.73)	4887	49.64 (49.61 to 49.67)	3.33 (3.30 to 3.36)
**Southeast**					
25–29	363	0.81 (0.80 to 0.81)	5508	12.37 (12.37 to 12.37)	1.61 (1.59 to 1.62)
30–34	772	1.82 (1.81 to 1.83)	9593	22.70 (22.69 to 22.71)	2.71 (2.68 to 2.73)
35–39	1244	3.13 (3.13 to 3.14)	14,480	36.46 (36.44 to 36.47)	3.36 (3.33 to 3.38)
40–44	1961	5.35 (5.34 to 5.35)	20,572	56.13 (56.11 to 56.14)	3.83 (3.80 to 3.86)
45–49	2460	7.56 (7.55 to 7.57)	21,140	64.99 (64.97 to 65.01)	3.67 (3.64 to 3.70)
50–54	2576	9.43 (9.42 to 9.43)	15,212	55.70 (55.67 to 55.72)	3.33 (3.31 to 3.34)
55–59	2380	10.99 (10.98 to 10.99)	10,837	55.05 (50.02 to 50.08)	2.91 (2.89 to 2.91)
60–64	2242	12.70 (12.69 to 12.71)	9223	52.27 (52.24 to 52.30)	2.57 (2.55 to 2.58)
**South**					
25–29	208	1.43 (1.42 to 1.43)	4341	29.93 (29.91 to 29.95)	2.60 (2.55 to 2.66)
30–34	414	2.96 (2.95 to 2.97)	6679	47.81 (47.78 to 47.84)	4.39 (4.31 to 4.47)
35–39	715	5.29 (5.28 to 5.29)	9998	73.98 (73.95 to 74.01)	5.53 (5.44 to 5.61)
40–44	893	7.10 (7.09 to 7.10)	14,092	112.17 (112.11 to 112.23)	5.24 (5.18 to 5.30)
45–49	1115	9.90 (9.89 to 9.90)	15,274	135.64 (135.57 to 135.71)	4.95 (4.89 to 5.00)
50–54	1073	11.45 (11.44 to 11.45)	9969	106.41 (106.35 to 106.48)	4.02 (3.99 to 4.05)
55–59	996	13.19 (13.18 to 13.19)	5722	75.83 (75.77 to 75.88)	3.36 (3.34 to 3.39)
60–64	898	14.75 (14.73 to 14.76)	4390	72.14 (72.08 to 72.19)	2.76 (2.73 to 2.78)
**Central-West**					
25–29	101	1.26 (1.26 to 1.27)	1155	14.41 (14.41 to 14.41)	2.40 (2.33 to 2.47)
30–34	211	2.82 (2.81 to 2.82)	2167	29.02 (28.99 to 29.04)	3.99 (3.89 to 4.08)
35–39	310	4.65 (4.64 to 4.66)	3503	52.58 (52.55 to 52.61)	4.56 (4.47 to 4.66)
40–44	429	7.50 (7.49 to 7.50)	4706	82.33 (82.26 to 82.40)	5.43 (5.32 to 5.54)
45–49	546	11.55 (11.53 to 11.57)	4255	90.02 (89.94 to 90.10)	5.41 (5.32 to 5.51)
50–54	511	13.72 (13.70 to 13.73)	2565	68.87 (68.81 to 68.93)	4.39 (4.32 to 4.46)
55–59	467	16.30 (16.28 to 16.32)	1555	54.29 (54.23 to 54.35)	3.76 (3.71 to 3.82)
60–64	397	17.93 (17.90 to 17.96)	1269	57.33 (57.25 to 57.41)	3.06 (3.02 to 3.10)
**Brazil**					
25–29	1196	1.14 (1.13 to 1.15)	15,396	14.79 (14.79 to 14.79)	2.40 (2.39 to 2.42)
30–34	2451	2.54 (2.53 to 2.54)	27,232	28.25 (28.24 to 28.25)	3.99 (3.96 to 4.02)
35–39	3848	4.35 (4.34 to 4.36)	42,535	48.12 (48.11 to 48.13)	4.57 (4.54 to 4.59)
40–44	5382	6.76 (6.75 to 6.76)	58,413	73.40 (73.38 to 73.41)	5.43 (5.40 to 5.45)
45–49	6674	9.63 (9.63 to 9.64)	58,507	84.45 (84.44 to 84.47)	5.41 (5.38 to 5.43)
50–54	6583	11.44 (11.43 to 11.45)	38,866	67.58 (67.56 to 67.60)	4.40 (4.38 to 4.41)
55–59	6143	13.33 (13.32 to 13.33)	25,339	55.01 (55.00 to 55.03)	3.77 (3.75 to 3.78)
60–64	5615	14.91 (14.90 to 14.91)	20,687	54.95 (54.94 to 54.97)	3.07 (3.06 to 3.08)

* Crude rate. International classification of diseases. 10th revision. Codes C53. Source: Mortality Information System (SIM) and Hospital Information System (SIH/SUS). Data made available by the Department of Informatics of The National Health System (DATASUS—www.datasus.gov.br (accessed on 12 April 2019). Ministry of Health. Brazil.

**Table 2 ijerph-18-10966-t002:** Mortality linear regression * and incidence * of hospitalization (per 100,000 inhabitants) between 2000 and 2012, according to the regions of the country.

Country Regions	Linear Regression Mortality			Linear Regression Admissions		
	β	*p* *	r^2^	β	*p* *	r^2^
**Age group (years) ***						
**North**						
25–29	0.03	0.38	0.07	−0.27	0.135	0.19
30–34	0.12	0.148	0.18	−0.72	0.025	0.37
35–39	0.17	0.229	0.12	−2.14	0.001	0.63
40–44	0.44	0.036	0.34	−2.45	0.026	0.38
45–49	0.83	0.015	0.43	−2.58	0.031	0.36
50–54	0.28	0.246	0.12	0.09	0.926	0.01
55–59	0.80	0.071	0.26	1.23	0.108	0.21
60–64	0.67	0.014	0.43	1.71	0.072	0.26
**Northeast**						
25–29	0.05	0.031	0.35	−0.05	0.838	0.01
30–34	0.11	0.001	0.64	−0.56	0.193	0.15
35–39	0.12	0.039	0.33	−1.64	0.048	0.30
40–44	0.05	0.241	0.12	−2.30	0.078	0.26
45–49	0.19	0.007	0.49	−2.56	0.124	0.20
50–54	0.23	0.150	0.17	−2.06	0.046	0.31
55–59	0.38	0.014	0.43	−1.20	0.060	0.29
60–64	0.29	0.011	0.46	−1.13	0.059	0.28
**Southeast**						
25–29	0.01	0.088	0.24	−0.08	0.692	0.02
30–34	0.03	0.108	0.22	−0.40	0.027	0.37
35–39	−0.03	0.191	0.15	−1.62	<0.001	0.89
40–44	−0.12	0.001	0.65	−3.23	<0.001	0.94
45–49	−0.26	<0.001	0.72	−4.03	<0.001	0.88
50–54	−0.30	0.007	0.49	−2.95	<0.001	0.82
55–59	−0.38	<0.001	0.81	−2.33	<0.001	0.78
60–64	−0.56	<0.001	0.85	−2.31	<0.001	0.84
**South**						
25–29	0.03	0.350	0.08	−0.34	0.656	0.02
30–34	0.01	0.836	0.04	−0.44	0.546	0.03
35–39	−0.01	0.894	0.01	−2.21	0.066	0.27
40–44	−0.28	0.001	0.62	−5.90	0.024	0.38
45–49	−0.49	<0.001	0.84	−8.87	0.025	0.37
50–54	−0.59	0.001	0.63	−6.26	0.045	0.32
55–59	−0.53	0.001	0.68	−3.89	0.014	0.44
60–64	−0.39	0.003	0.56	−1.43	0.236	0.12
**Central-west**						
25–29	−0.01	0.814	0.01	−0.47	0.008	0.48
30–34	0.10	0.088	0.24	−1.37	<0.001	0.80
35–39	0.08	0.188	0.15	−3.15	<0.001	0.79
40–44	−0.02	0.790	0.01	−4.55	0.001	0.68
45–49	−0.22	0.381	0.07	−6.07	<0.001	0.79
50–54	−0.11	0.650	0.01	−3.21	<0.001	0.85
55–59	−0.36	0.131	0.19	−1.82	0.004	0.54
60–64	−0.36	0.333	0.08	−1.12	0.159	0.18
**Brazil**						
25–29	0.03	0.006	0.50	−0.16	0.505	0.04
30–34	0.06	0.001	0.64	−0.58	0.047	0.31
35–39	0.03	0.115	0.21	−1.89	<0.001	0.71
40–44	−0.05	0.092	0.23	−3.43	0.001	0.68
45–49	−0.11	0.087	0.24	−4.49	0.001	0.63
50–54	−0.16	0.023	0.39	−3.14	0.003	0.57
55–59	−0.15	0.050	0.31	−2.08	0.002	0.60
60–64	−0.24	0.001	0.66	−1.57	0.001	0.64

* Crude rate. International classification of diseases, 10th revision. Codes C53. β: regression slope; r^2^: predictive capacity; 95% CI: 95% confidence interval. Source: Mortality Information System (SIM) and Hospital Information System (SIH/SUS). Data made available by the Department of Informatics of The National Health System (DATASUS—www.datasus.gov.br (accessed on 12 April 2019)). Ministry of Health, Brazil.

**Table 3 ijerph-18-10966-t003:** Mortality * and incidence * of hospitalization per 100,000 inhabitants (95% confidence interval), and linear regression estimate between 2000 and 2012 according to the regions of the country.

**Brazil/Regions**	**Age-Standardized Mortality * (95% CI) by Cervical Cancer † (×100,000 Inhabitants)**													**Linear Regression**		
	**2000**	**2001**	**2002**	**2003**	**2004**	**2005**	**2006**	**2007**	**2008**	**2009**	**2010**	**2011**	**2012**	**β**	* **p** *	**r^2^**
North	4.09 (4.08; 4.10)	5.20 (5.19; 5.20)	6.28 (6.27; 6.29)	6.21 (6.21; 6.22)	7.32 (7.31; 7.33)	7.07 (7.06; 7.08)	7.40 (7.39; 7.41)	5.84 (5.83; 5.85)	7.22 (7.21; 7.22)	7.38 (7.37; 7.38)	6.88 (6.87; 6.88)	6.93 (6.92; 6.93)	7.39 (7.38; 7.39)	0.18	0.008	0.48
Northeast	2.85 (2.84; 2.86)	3.10 (3.09; 3.10)	3.40 (3.39; 3.40)	3.28 (3.27; 3.29)	3.61 (3.60; 3.61)	3.92 (3.92; 3.93)	4.07 (4.06; 4.07)	3.75 (3.74; 3.75)	3.84 (3.83; 3.84)	4.00 (3.99; 4.00)	3.62 (3.61; 3.62)	3.73 (3.72; 3.74)	4.12 (4.12; 4.13)	0.07	0.002	0.61
Southeast	3.17 (3.16; 3.17)	3.21 (3.21; 3.22)	2.97 (2.96; 2.97)	3.04 (3.04; 3.05)	3.28 (3.27; 3.28)	2.83 (2.82; 2.83)	2.69 (2.69; 2.70)	2.46 (2.46; 2.47)	2.44 (2.44; 2.45)	2.37 (2.36; 2.37)	2.47 (2.46; 2.48)	2.51 (2.51; 2.52)	2.41 (2.41; 2.42)	−0.07	<0.001	0.79
South	4.03 (4.02; 4.04)	4.62 (4.61; 4.62)	3.95 (3.95; 3.96)	4.08 (4.08; 4.09)	3.99 (3.98; 3.99)	3.78 (3.78; 3.79)	3.59 (3.58; 3.59)	3.14 (3.13; 3.15)	3.27 (3.26; 3.28)	3.10 (3.09; 3.10)	3.02 (3.01; 3.02)	3.14 (3.13; 3.14)	3.25 (3.25; 3.25)	−0.11	<0.001	0.79
Central-west	4.03 (4.02; 4.04)	3.81 (3.80; 3.81)	3.50 (3.49; 3.51)	4.57 (4.57; 4.58)	4.61 (4.61; 4.61)	4.68 (4.67; 4.69)	4.53 (4.53; 4.54)	3.89 (3.89; 3.90)	3.81 (3.81; 3.81)	3.89 (3.89; 3.89)	4.04 (4.03; 4.05)	3.49 (3.48; 3.49)	3.59 (3.58; 3.59)	−0.03	0.270	0.10
Brazil	3.34 (3.33; 3.35)	3.57 (3.57; 3.58)	3.45 (3.45; 3.45)	3.55 (3.55; 3.56)	3.78 (3.77; 3.78)	3.62 (3.61; 3.63)	3.56 (3.55; 3.56)	3.17 (3.17; 3.17)	3.29 (3.29; 3.30)	3.28 (3.27; 3.28)	3.22 (3.21; 3.23)	3.26 (3.25; 3.26)	3.36 (3.35; 3.37)	−0.02	0.065	0.27
**Brazil/Regions**	**Age-Standardized Incidence * (95% CI) of Hospital Admissions for Cervical Câncer (× 100,000 Inhabitants)**													**Linear Regression**		
	**2000**	**2001**	**2002**	**2003**	**2004**	**2005**	**2006**	**2007**	**2008**	**2009**	**2010**	**2011**	**2012**	**β**	** *p* **	**r^2^**
North	17.56 (17.54; 17.58)	18.32 (18.29; 18.35)	20.39 (20.37; 20.41)	24.03 (24.01; 24.06)	28.24 (28.20; 28.280	25.64 (25.61; 25.68)	22.51 (22.49; 22.53)	22.03 (22.01; 22.06)	20.02 (19.99; 20.04)	18.36 (18.35; 18.37)	15.49 (15.47; 15.50)	15.99 (15.98; 16.01)	16.13 (16.12; 16.14)	−0.42	0.157	0.17
Northeast	24.60 (24.59; 24.61)	21.37 (21.36; 21.39)	29.20 (29.19; 29.21)	37.77 (37.75; 37.80)	32.87 (32.86; 32.89)	27.79 (27.77; 27.81)	29.22 (29.21; 29.23)	26.10 (26.08; 26.11)	25.70 (25.68; 25.71)	23.51 (23.49; 23.53)	20.72 (20.71; 20.73)	20.34 (20.32; 20.35)	20.33 (20.32; 20.35)	−0.67	0.080	0.25
Southeast	24.08 (24.06; 24.09)	24.16 (24.15; 24.18)	26.24 (26.23; 26.25)	25.18 (25.16; 25.19)	22.59 (22.57; 22.60)	22.84 (22.82; 22.85)	21.94 (21.94; 21.95)	19.00 (18.99; 19.00)	17.10 (17.09; 17.10)	16.41 (16.40; 16.42)	16.93 (16.93; 16.94)	15.04 (15.04; 15.05)	15.62 (15.61; 15.62)	−0.95	<0.001	0.88
South	33.98 (33.95; 34.01)	30.81 (30.79; 30.84)	53.69 (53.65; 53.73)	54.81 (54.76; 54.86)	53.45 (53.40; 53.50)	55.37 (55.32; 55.42)	50.34 (50.30; 50.37)	40.54 (40.51; 40.57)	31.93 (31.90; 31.95)	29.82 (29.79; 29.84)	28.32 (28.29; 28,35)	26.45 (26.44; 26.46)	24.78 (24.76; 24.79)	−1.43	0.236	0.12
Central-west	36.96 (36.92; 37.00)	36.72 (36.67; 36.77)	28.04 (28.00; 28.08)	31.21 (31.24; 31.24)	28.38 (28.34; 28.41)	27.09 (27.06; 27.13)	29.31 (29.27; 29.34)	22.09 (22.06; 22.11)	24.13 (24.10; 24.15)	24.99 (24.97; 25.02)	22.70 (22.67; 22.72)	20.17 (20.15; 20.19)	19.34 (19.32; 19.36)	−1.33	<0.001	0.84
Brazil	26.31 (26.30; 26.32)	25.07 (25.06; 25.07)	31.16 (31.15; 31.17)	33.36 (33.35; 33.38)	30.75 (30.74; 30.75	29.65 (29.64; 29.65)	28,74 (28.74; 28.75)	24.51 (24.50; 24.52)	22.20 (22.19; 22.20)	20.97 (20.97; 20.97)	19.90 (19.89; 19.90)	18.54 (18.53; 18.54)	18.48 (18.48; 18.48)	−1.01	0.002	0.59

* Standardized for age according to the world population from the World Health Organization. † International classification of diseases, 10th revision. Codes C53. β: regression slope; r^2^: predictive capacity; 95% CI: 95% confidence interval. Source: Mortality Information System (SIM) and Hospital Information System (SIH/SUS). Data made available by the Department of Informatics of The National Health System (DATASUS—www.datasus.gov.br (accessed on 20 May 2019)). Ministry of Health, Brazil.

## Data Availability

The datasets used and/or analyzed during the current study are available from the corresponding author on reasonable request.

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
