# Peer review of "Inequalities in Mortality and Access to Hospital Care for Cervical Cancer—An Ecological Study"

_ijerph, 2021, doi:10.3390/ijerph182010966_

Round 1
Reviewer 1 Report
The fundamental problem of writing in English limits the capability to assess the scientific soundness of the study. There is a widespread lack of paragraphs with topic sentences followed by sentences with evidence supporting the main idea. Also the vocabulary is often misused, for example this is an epidemiological, not "ecological" study. Ecological means 1. (Environmental Science) of or relating to ecology or 2. (Environmental Science) (of a practice, policy, product, etc) tending to benefit or cause minimal damage to the environment. Hence, such a study investigates “the science of the relationships between organisms and their environments or b. the relationship between organisms and their environment. This study doesn’t investigate ecological relationships. Epidemiology is the method used to find the causes of health outcomes and diseases in populations. In epidemiology, the patient is the community and individuals are viewed collectively. By definition, epidemiology is the study (scientific, systematic, and data-driven) of the distribution (frequency, pattern) and determinants (causes, risk factors) of health-related states and events (not just diseases) in specified populations (neighborhood, school, city, state, country, global). It is also the application of this study to the control of health problems (Source: Principles of Epidemiology, 3rd Edition).
Translation software alone is not an appropriate method to write a scientific paper. The author’s institute has scholars in English Language and Technical English that can help revise this paper to be properly reviewed for its scientific content.
As for the data, it is difficult to track any trend in the data, especially when single entries are broken by column or rows. Tables are too dense to read the data and imagine the patterns. Also there are 4 quartiles (for example, 0-25%, 26-50%, 51-75%, 76-100% for first, second, third and fourth quartile respectively), not 3, as depicted in the geograpical maps of disease incidence.
Author Response
Response 1:
In Ecological Studies the measures used represent characteristics of population groups. Therefore, the unit of analysis is the population and not the individual. This study helps to identify factors that deserve a more detailed investigation through studies with greater analytical capacity.
In time series studies, in which the same area or population is studied at different times in time, they are classified as a subtype of ecological studies. In this case, each time unit would be treated as a complete ecological unit. Ecological studies are generally based on secondary data:
Disease data sources:
- a) Mortality records;
- b) Morbidity records;
- c) Census data on morbidity, mortality and population.
Sources of exposure data and confounding factors:
- a) Economic censuses;
- b) Demographic censuses;
- c) Production and/or consumption data.
CARVALHO, Eduardo Rebouças; ROCHA, Hermano Alexandre Lima. Estudos epidemiológicos. Faculdade de, 2008.
MENEZES, Ana; SANTOS, Iná da S. dos. Curso de epidemiologia básica para pneumologistas. 4ª parte-Epidemiologia clínica. Jornal de Pneumologia, v. 25, p. 321-326, 1999.
DA SILVA PAIVA, Laércio et al. Decline in Stroke Mortality Between 1997 and 2012 by Sex: Ecological Study in Brazilians Aged 15 to 49 Years. Scientific reports, v. 9, n. 1, p. 1-8, 2019.
NUNES, Michele Ribeiro Alexandre; SOUSA, Luiz Vinicius de Alcantara; NASCIMENTO, Vânia Barbosa do. Mortalidade infantil na Região Metropolitana de São Paulo: estudo ecológico. Einstein (São Paulo), v. 19, 2021.
ALCANTARA, Stefanie de Sousa Antunes et al. Epidemiological profile of prostate cancer mortality and access to hospital care in Brazilian regions-an ecological study. Journal of Human Growth and Development, v. 31, n. 2, p. 310-317, 2021.
The study underwent a review of English, follows a new version and certificate.
Response 2:
In the figures mentioned by the reviewer, only the 3 quartiles are listed.

Reviewer 2 Report
This article is very interesting for the incidence of cervical cancer in different regions of Brazil. It is interesting that the incidence of this type of cancer depends on many factors, such as the region of the country, age, economic situation and education. The same can be seen in other countries.
The authors carefully analyzed patients aged 25-64 in the years 2000-2012, focusing on their mortality and hospitalization.
I feel to addressed the following minor points:
Point 1: Line 67-70. It is: “The distribution of cervical cancer admission and mortality rates according to the topographic location of the uterine lesion coded according to the 10th International Classification of Diseases (ICD) [10] was analyzed:
- C53 - Malignant neoplasm of cervix;”
and it should be: “The distribution of cervical cancer admission and mortality rates according to the topographic location of the uterine lesion coded according to the 10th International Classification of Diseases (ICD) [10] was analyzed: C53 - Malignant neoplasm of cervix.” (unless, this sentence is not complete)
Point 2: Line 94. Replce: „2.4Cervical cancer mortality” with „2.4 Cervical cancer mortality” (Style)
Point 3: Line 149-157. Replace: „It was 149 observed that only North and Northeast had an increase in mortality (β = 0.18, p = 0.008, 150 r2 = 0.48, β = 0.07, p = 0.002, r2 = 0.61, respectively). with the other regions presenting a reduction, Southeast (β = -0.07, p <0.001, r2 = 0.79), South (β = -0.11, p <0.001, r2 = 0.79) and Central West (β = -0.03, p = 0.008, r2 = 0.48). During the years 2000 to 2012, the greatest 153 reduction in the incidence of UCC was in the South Region (β = -1.43, p = 0.236, r2 = 0.12), followed by the Center-West (β = -1 , P <0.001, r2 = 0.84), Southeast (β = -0.95, p <0.001, r2 = 155 0.88), Northeast (β = -0.67, p = 0.080, r2 = 0 , 25) and, finally, North (β = -0.42, p = 0.157, r2 = 156 0.17) (Figure 3 and figure 4). With : „It was observed that only North and Northeast had an increase in mortality (β = 0.18, p = 0.008, r2 = 0.48, β = 0.18, p = 0.002, r2 = 0.48, respectively) with the other regions presenting a reduction, Southeast (β = -0.07, p <0.001, r2 = 0.79), South (β = -0.11, p <0.001, r2 = 0.79) and Central West (β = -0.03, p = 0.278, r2 = 0.1) (Figure 3). During the years 2000 to 2012, the greatest reduction in the incidence of UCC was in the South Region (β = -1.43, p = 0.236, r2 = 0.12), followed by the Center-West (β = -1.01 ,p <0.002, r2 = 0.59), Southeast (β = -0.95, p <0.001, r2 = 155 0.88), Northeast (β = -0.67, p = 0.080, r2 = 0 , 25) and finally North (β = -0.42, p = 0.157, r2 = 156 0.17) (Figure 4).”
Point 4: Line 166-167. Why in the entire article are patients in the age range from 25 to 64, and Figure 3 and Figure 4 are patients in the age range from 15 to 64?
Point 5: In Table 3, I would change the font in the Linear Regression column for better visibility. (if it's possible).

Author Response
Point 1: Line 67-70. It is: “The distribution of cervical cancer admission and mortality rates according to the topographic location of the uterine lesion coded according to the 10th International Classification of Diseases (ICD) [10] was analyzed:
- C53 - Malignant neoplasm of cervix;”
and it should be: “The distribution of cervical cancer admission and mortality rates according to the topographic location of the uterine lesion coded according to the 10th International Classification of Diseases (ICD) [10] was analyzed: C53 - Malignant neoplasm of cervix.” (unless, this sentence is not complete)
Response 1:
Suggestion accepted and changed in the document.
Line 67-69: “The distributions of cervical cancer admissions and mortality rates were coded according to the 10th International Classification of Diseases (ICD) as C53—malignant neoplasm of the cervix based on the topographic locations of the uterine lesions analyzed [10].”
Point 2: Line 94. Replce: „2.4Cervical cancer mortality” with „2.4 Cervical cancer mortality” (Style)
Response 2:
Suggestion accepted and changed in the document.
Line 94: “Cervical cancer mortality”
Point 3: Line 149-157. Replace: „It was 149 observed that only North and Northeast had an increase in mortality (β = 0.18, p = 0.008, 150 r2 = 0.48, β = 0.07, p = 0.002, r2 = 0.61, respectively). with the other regions presenting a reduction, Southeast (β = -0.07, p <0.001, r2 = 0.79), South (β = -0.11, p <0.001, r2 = 0.79) and Central West (β = -0.03, p = 0.008, r2 = 0.48). During the years 2000 to 2012, the greatest 153 reduction in the incidence of UCC was in the South Region (β = -1.43, p = 0.236, r2 = 0.12), followed by the Center-West (β = -1 , P <0.001, r2 = 0.84), Southeast (β = -0.95, p <0.001, r2 = 155 0.88), Northeast (β = -0.67, p = 0.080, r2 = 0 , 25) and, finally, North (β = -0.42, p = 0.157, r2 = 156 0.17) (Figure 3 and figure 4). With : „It was observed that only North and Northeast had an increase in mortality (β = 0.18, p = 0.008, r2 = 0.48, β = 0.18, p = 0.002, r2 = 0.48, respectively) with the other regions presenting a reduction, Southeast (β = -0.07, p <0.001, r2 = 0.79), South (β = -0.11, p <0.001, r2 = 0.79) and Central West (β = -0.03, p = 0.278, r2 = 0.1) (Figure 3). During the years 2000 to 2012, the greatest reduction in the incidence of UCC was in the South Region (β = -1.43, p = 0.236, r2 = 0.12), followed by the Center-West (β = -1.01 ,p <0.002, r2 = 0.59), Southeast (β = -0.95, p <0.001, r2 = 155 0.88), Northeast (β = -0.67, p = 0.080, r2 = 0 , 25) and finally North (β = -0.42, p = 0.157, r2 = 156 0.17) (Figure 4).”
Response 3:
Suggestion accepted and changed in the document.
Line 148-157: Variations in mortality and hospitalization rates by Brazilian region can be visualized in figure 1 and figure 2, which present the quartiles for the observed periods. It was observed that only the North and the Northeast experienced increases in mortality (β = 0.18, p = 0.008, r2 = 0.48; β = 0.18, p = 0.002, r2 = 0.48, respectively), and the other regions presented reduced rates: Southeast (β = -0.07, p <0.001, r2 = 0.79), South (β = -0.11, p <0.001, r2 = 0.79), and Central-West (β = -0.03, p = 0.278, r2 = 0.1) (Figure 3). Between 2000 and 2012, the greatest reduction in the incidence of UCC occurred in the South region (β = -1.43, p = 0.236, r2 = 0.12) followed by the Central-West (β = -1.01, p <0.002, r2 = 0.59), the Southeast (β = -0.95, p <0.001, r2 =0.88), the Northeast (β = -0.67, p = 0.080, r2 = 0.25), and, finally, the North (β = -0.42, p = 0.157, r2 =0.17) (Figure 4).
Point 4: Line 166-167. Why in the entire article are patients in the age range from 25 to 64, and Figure 3 and Figure 4 are patients in the age range from 15 to 64?
Response 4:
Adjusted.
Line 167: Figure 3. Incidence of hospital admissions for cervical câncer (x100,000 inhabitants), resident in Brazil aged 25 - 64 years, in the period from 2000 to 2012 and estimates obtained from linear regression, according to year and regions.
Line 170: Figure 4. Mortality rate for cervical cancer (100,000 inhabitants), resident in Brazil aged 25 - 64 years, in the period from 2000 to 2012 and estimates obtained from linear regression, according to year and regions.
Point 5: In Table 3, I would change the font in the Linear Regression column for better visibility. (if it's possible).
Response 5:
Suggestion accepted and changed in the in table 3.
Thanks for your review.

Reviewer 3 Report
Reviewer statement:
Inequalities in Mortality and Access to Hospital Care for Cervical Cancer
Cervical cancer is considered one of the most common form of cancer in women worldwide and one of the leading causes of cancer deaths in women. The authors performed a population-based ecological study between 2000 and 2012, which is excellent. This knowledge can provide valuable information and guidance for the management and especially prevention of cervical cancer. This information could guide treatment, further knowledge and research on this topic.
Title: The title reflects the topic being investigated. The article type could be added to the title.
Overall: The paper is well written, the English grammar is considered adequate.
Abstract : see overall remarks and remarks throughout the article.
Introduction:
The introduction section is attractive to read form a reader point of view, explaining the reason for conducting this study.
Methods :
This section is attractive to read form a reader point of view, explaining the methods used to conduct this study.
- On page the authors report stratifying the Brazilian administrative regions. This should be explained to the reader why the authors choose to stratify according to administrative regions.
- The deaths recorded by the cervical cancer information system (SIM) were collected. The question remains if women were eligible to participate in cervical screening ?
- One of the most important questions from a reader perspective, when was the cervical cancer defined as ill=defined cause, the reason for mortality. If women had more than one possible reason, how was this coded in the system?
- In line 108 the word UCC should be replaced by CCU.
Results:
This section was easy to read. The result are well presented.
No comments on this paragraph.
Discussion
The discussion section is easy to read. Despite, there are some important questions.
- The authors report significant differences in hospitalization and deaths in age groups and according to federative regions. The authors should try bot provide reasons for the reported results. This is crucial and essential as this might lead to important knowledge and possibilities to improve the reported outcomes.
- The question remains if the reported differences can be explained by difference in social economic levels, please elucidate on this point.
Tables:
No comments
Author Response
Point 1: The title reflects the topic being investigated. The article type could be added to the title.
Response 1:
Suggestion accepted and changed in the document.
Line 2: “Inequalities in Mortality and Access to Hospital Care for Cervical Cancer—An Ecological study”
Point 2: Methods
This section is attractive to read form a reader point of view, explaining the methods used to conduct this study.
- On page the authors report stratifying the Brazilian administrative regions. This should be explained to the reader why the authors choose to stratify according to administrative regions.
- The deaths recorded by the cervical cancer information system (SIM) were collected. The question remains if women were eligible to participate in cervical screening ?
- One of the most important questions from a reader perspective, when was the cervical cancer defined as ill=defined cause, the reason for mortality. If women had more than one possible reason, how was this coded in the system?
- In line 108 the word UCC should be replaced by CCU.
Response 2:
- Line 51-53: This is an ecological, population-based study that uses Brazilian data to evaluate temporal trends in cervical cancer between the years 2000 and 2012. Brazilian admin-istrative regions were chosen as study areas in order to ensure more complete data.
- Line 55-62: The survey is composed of all deaths from cervical cancer registered by the Cervi-cal Cancer Information System (SIM) in women between 25 and 64 years of age, the age group in which a cytopathological exam (Pap smear) is offered to Brazilian wom-en. The Pap smear is used for screening and diagnosis of CC [5,6]. The specific period studied ranged from January 1, 2000, to December 31, 2012. Data from the Department of Informatics of the Unified Health System (DATASUS—www.datasus.gov.br), a free access database that represents the main source of health information in the country, provide health information for states, municipalities, and the Federal District [4].
- The mortality system uses the underlying cause of death (ICD-10) to record death, so we have a limitation because we were unable to identify the secondary causes that led to death.
- Line 7: Suggestion accepted and changed in the document.
Point 3: Discussion
The discussion section is easy to read. Despite, there are some important questions.
- The authors report significant differences in hospitalization and deaths in age groups and according to federative regions. The authors should try bot provide reasons for the reported results. This is crucial and essential as this might lead to important knowledge and possibilities to improve the reported outcomes.
- The question remains if the reported differences can be explained by difference in social economic levels, please elucidate on this point.
Response 3:
Line 195-199: In Brazil, inequality in access to health services is still frequently discussed due to its influence on the high incidence, diagnosis in more advanced stages, and high mor-tality rate of CC. This supports the theory that even though the number of preventive exams have increased in recent years, access to health services is still related to socio-economic levels within the country
Thanks for your review.

Round 2
Reviewer 3 Report
The authors have adressed the raised points adequately, hereby improving the quality of the article. This makes the articel ready for publication